# Nutrient Uptake of Two Semidomesticated *Jaltomata* Schltdl. Species for Their Cultivation

**DOI:** 10.3390/plants14071124

**Published:** 2025-04-04

**Authors:** Ignacio Darío Flores-Sánchez, Manuel Sandoval-Villa, Ebandro Uscanga-Mortera

**Affiliations:** 1Postgraduate in Edaphology, Colegio de Postgraduados, Mexico-Texcoco Highway, km 36.5, Montecillo, Texcoco 56264, Mexico; msandoval@colpos.mx; 2Postgraduate in Botany, Colegio de Postgraduados, Mexico-Texcoco Highway, km 36.5, Montecillo, Texcoco 56264, Mexico; euscanga@colpos.mx

**Keywords:** native plant resources, absorption curves, protected agriculture, hydroponics

## Abstract

The nutrient uptake of a species under cultivated conditions is important for program fertilization. The *Jaltomata* genus has two semidomesticated species, *J. procumbens* and *J. tlaxcala*, used as food and considered with potential for their study in controlled environments. The objective of this research was to determine nutrient uptake curves of these species in a greenhouse and using hydroponics. The research was carried out at the Colegio de Postgraduados, Campus Montecillo, Texcoco, State of Mexico, from August to November 2020. The treatments included the following: two species and three electrical conductivity levels: 1, 2, and 3 dS m^−1^. Nutrients in leaf and total dry matter (TDM) were determined. Variability between species and phenological stages on the nutrient concentration and accumulation of TDM was observed. For macronutrients, *J. procumbens* concentrated in descending order more P from the vegetative stage (4.21–2.43 g kg^−1^ dry matter), and Mg until fructification (4.92–3.26 g kg^−1^ dry matter), for K it was higher at vegetative (52.29 g kg^−1^ dry matter) and harvesting stages (26.05 g kg^−1^ dry matter), and N (23.92 g kg^−1^ dry matter) at flowering; *J. tlaxcala* concentrated more Ca from fructification (10.10–13.85 g kg^−1^ dry matter). For micronutrients, *J. tlaxcala* concentrated more Fe from the vegetative stage (157.7–207.5 mg kg^−1^ dry matter), B and Zn at 23.3–38.4 and 26.04–28.45 mg kg^−1^ dry matter, respectively, from flowering, and Mn (108.4–232.28 mg kg^−1^ dry matter) from fructification. The main structures of TDM accumulation by vegetative stage in *J. procumbens* were the leaf and root (vegetative and flowering), root and stem (fructification), and reproductive structures and root (harvesting); in *J. tlaxcala*, the main structures were the leaf and root (vegetative), root and leaf (flowering and fructification), and root and reproductive structures (harvesting). Due to this variability, specific fertilization programs are required for each species.

## 1. Introduction

Plant nutrition is one of the most important aspects of crop production [1], since optimal growth and yield will depend on it [2].

The genus *Jaltomata* Schltdl., family Solanaceae, has two recognized diversity centers, South America and Mexico with 63 and 8 species, respectively. They are herbaceous and shrubby with edible fruits [3]. In Mexico, *J. procumbens* of erect growth and blue dark fruit is found as ruderal and arvense (plant located on the edge or inside of crop fields useful for human being); *J. tlaxcala*, recently recognized as a new species [4], is located in specific regions of the country, has a decumbent growth habit with green fruit in the mature stage, and is found as an arvense. They are considered a semidomesticated species and are linked to traditional food production systems in Mexico, characterized by being on hillsides or superficial soils, and with frequently degraded soils [5]. Around 64% of Mexican agricultural lands present a level of degradation, and 28% of farmers report some problem with low fertility of their soils [6]. The fruits of these species are recollected and consumed fresh or after some preparations [7].

Their study is relevant due to their important nutrient content and nutraceutical compounds [8,9,10], characteristics that could help address the problem of malnutrition mentioned by the World Health Organization [11], which is linked to the reduced diversity of foods in production systems and by the poor supply of vitamins, micronutrients, and nutraceutical compounds, because 60% of the 90% of calories consumed worldwide by the population are provided by the main basic crops rich in carbohydrates (rice, corn, potato, soybeans, and wheat) [12,13,14,15].

Also, these species are considered viable for their future use because these meet the criteria to consider new plant resources for food purposes [16]: edible plants for human being, plants with high nutritional value; plants that do not need intermediate process between growing, harvest, and use; and plants preferentially native to Mexico.

From the evaluation of agronomic traits, both species have the potential to be studied in controlled environments [7]; however, based on the literature review, there is still no information on the demand and nutrient uptake for both species.

The dynamic uptake of nutrients by the crop is important, to generate the absorption curves of nutrients for each phenological stage, which allows the identification of the maximum absorption stages during the growing season and for the design of fertilization programs [17].

These curves are determined by genetic factors, phenological stage, and weather conditions [1]; then, it is necessary to determine the nutrient uptake dynamics of *Jaltomata procumbens* and *J. tlaxcala*, according to their nutrimental demand during the growing season, which allows their management under greenhouse and hydroponic conditions, which are different from those present under their natural environment, considering the hypothesis that each species has a different dynamics uptake of macro and micronutrients in each phenological stage.

That is why the objective of this research was to determine the absorption curves of nutrients for both *Jaltomata* species for the vegetative, flowering, fructification, and harvesting stages in greenhouse and hydroponic conditions.

## 2. Results

### 2.1. Analysis of Effects on the Nutrient Concentration and Total Dry Matter

The species (Sps) and electrical conductivity (EC) influenced the concentration of macronutrients and micronutrients in the leaf and the total dry matter (TDM), in at least one of the samplings, except for Mo and SPAD readings for Sps and B for EC (*p* ≤ 0.05). In the Sps x EC interaction, in at least one of the samplings, statistical differences were registered in macronutrients, micronutrients, and TDM and SPAD readings, which indicated that the EC effect for each variable was not the same within each species. The highest coefficients of variation were recorded in micronutrients and TDM: TDM (21.1%), B (41.2%), Mo (42.5%), and Zn (23.7%) for the first sampling; Fe (25.8%) and Mo (48.8%) in the second sampling; TDM (29.6%) and Mo (53.0%) third sampling; and TDM (32.5%), Fe (29.7%) and Mo (73.4%) for the fourth sampling.

### 2.2. Macronutrients

For N, except for the vegetative stage, there were statistical differences. In the flowering stage, an increase in both Sps was registered, greater in *J. procumbens* in 10.7% compared to *J. tlaxcala*. Subsequently, there was a reduction in both Sps, with *J. procumbens* maintaining a higher concentration in the fructification and harvest stages than *J. tlaxcala* (Figure 1). For P, the highest absorption was registered in *J. procumbens*, with statistical differences in the vegetative and fructification stages. The uptake dynamic was the same for both Sps, with the highest values in the first stage, gradually reducing until harvest, with the lowest values. For K, except for the flowering stage, there were statistical differences; in the vegetative stage, *J. procumbens* recorded more K than *J. tlaxcala*; from this stage, the uptake was reduced in both Sps; however, it was higher in 50.2% in *J. procumbens* at harvest. For *J. tlaxcala*, this reduction was less accentuated with 23.9%.

The Ca uptake increased in each of the phenological stages. The *J. tlaxcala* species statistically exceeded *J. procumbens* at harvest. The highest uptake occurred from fructification, *J. tlaxcala* ended with 21.2% more than *J. procumbens*. For Mg, statistical differences were presented in the vegetative, flowering, and harvest stages; in both Sps, the highest uptake occurred in the first stage, presenting *J. procumbens* the highest value, higher than *J. tlaxcala* by 19.6%; subsequently, a reduction was registered until the last stage, becoming more accentuated in *J. procumbens* with the lowest value compared to *J. tlaxcala*, by 38.5% (Figure 1).

### 2.3. Micronutrients

For B, there were statistical differences from the flowering stage, with 18.2 mg kg^−1^ for *J. procumbens* and 23.3 mg kg^−1^ for *J. tlaxcala*; subsequently, there was a gradual increase until the harvest stage, where *J. tlaxcala* obtained the highest concentration, higher than *J. procumbens* by 16.6% (Figure 2). For Fe, there were statistical differences in the vegetative stage where *J. tlaxcala* absorbed more Fe than *J. procumbens*, by 30.2%; subsequently, it reduced in both Sps to its lowest point in the flowering and fructification stages; from these stages, there was an increase where *J. tlaxcala* absorbed more Fe than *J. procumbens* by 9.6% for the harvest. The absorption of Mn had a constant increase with statistical differences in the vegetative and harvest stages, registering the highest concentration from fructification at the beginning, with the highest value coming from *J. tlaxcala* in the harvest, by 45.3% more than *J. procumbens*.

For Mo, *J. procumbens* registered the highest amount of this element, with the highest concentration in the flowering stage, higher by 38.0% than *J. tlaxcala*, which registered a decrease respect to the vegetative stage, although there were no statistical differences. For Zn, except for vegetative stage, there were statistical differences in the other stages, where *J. tlaxcala* registered a greater amount of this nutriment in flowering, fructification, and harvest, with respect to *J. procumbens* in 12.7, 12.4, and 18.7%, respectively (Figure 2).

### 2.4. Electrical Conductivity

Plant growth with 1 dS m^−1^ registered the highest concentration of Ca in the harvest stage, higher by 23.6% compared to 3 dS m^−1^ EC; for Mg, the highest concentration was recorded in the vegetative stage, with 15.4% more compared to 3 dS m^−1^ EC, indicating that the concentration decreases as EC is increased. For Mn, the highest concentration was registered in the fructification stage, higher by 19.9% compared to 3 dS m^−1^ EC (Table 1).

For plant growth with 2 dS m^−1^ EC, in the vegetative stage, the concentration of Ca was higher than 1 dS m^−1^ EC by 14.8 %; for fructification, the concentration of P, K, and Mo was higher by 23.8, 12.36, and 200% regarding 1 dS m^−1^ EC, respectively.

The plant growth with 3 dS m^−1^ EC presented higher values in the vegetative stage, with a higher concentration compared to 1 dS m^−1^ EC for P, 14.8%, and for Fe, 22.8%. In the flowering stage with this EC compared to the lowest EC, the highest concentration of N and Zn was obtained, standing out in 12.8 and 19.1%, respectively; for Mo, in this stage, a higher concentration was registered concerning 2 dS m^−1^ EC, higher by 57.9%. In fructification and harvest, for Zn, the highest concentration was presented with 3 dS m^−1^ EC, with the highest value in fructification, surpassing 1 dS m^−1^ EC by 40.9% (Table 1).

For SPAD readings, statistical differences were obtained in the fructification stage, registering the highest value with 2 dS m^−1^ EC (61.57), higher by 5.5% than 1 dS m^−1^ EC (Appendix A).

### 2.5. Sps × EC Combination

For the vegetative stage in the Sps x EC combination, an EC of 3 dS m^−1^ favored the P concentration in *J. procumbens*, for Ca and Fe in *J. tlaxcala* (Figure 3). The plant growth with 2 dS m^−1^ EC favored the P and Mg concentration in *J. tlaxcala*; this same EC level favored K, Ca, and Fe concentration in *J. procumbens*, while with 1 dS m^−1^ EC the highest value was registered for K and Mg, in *J. tlaxcala* and *J. procumbens*, respectively.

In the flowering stage, the plant growth with 3 dS m^−1^ EC favored the concentration of N, P, and Mo in both Sps. With 1 dS m^−1^ EC in both Sps, the highest concentration of Mg and Mn was registered. For B, 1 dS m^−1^ EC favored its concentration in *J. procumbens*, while in *J. tlaxcala* the highest value was registered with 3 dS m^−1^ EC (Figure 4).

For fructification, 1 dS m^−1^ EC favored the Ca and Mg concentration in *J. procumbens* and Mg in *J. tlaxcala*. With 2 dS m^−1^, the K concentration was favored in *J. procumbens*, while for *J. tlaxcala* the P and B concentration, with SPAD readings (63.17). For *J. procumbens*, with 3 dS m^−1^ EC the concentration of P, B, and Zn was promoted, with the SPAD readings (63.23), while, in *J. tlaxcala*, it favored K, Ca, and Zn concentration (Figure 5).

For the harvest stage, the plant growth with 1 dS m^−1^ EC favored the concentration of K, Ca, and Mg in *J. procumbens*, and in *J. tlaxcala* for Ca and Mg. With 2 dS m^−1^ EC, in *J. procumbens* the concentration of N was favored, and in *J. tlaxcala* for K and Mn. Finally, 3 dS m^−1^ EC helped a higher concentration of Mn and Zn in *J. procumbens*, while in *J. tlaxcala* in N and Zn (Figure 6).

### 2.6. Order of Macronutrients and Micronutrients Uptake

The macronutrients uptake in *J. tlaxcala* for the harvesting stage and, in both Sps, for the vegetative, flowering, and fructification stages was K > N > Ca > Mg > P, while for *J. procumbens* at harvesting was K > N > Ca > P > Mg. In the micronutrients, for both Sps, the order was Fe > Mn > B > Zn > Mo in the vegetative stage and, in flowering, Fe > Mn > Zn > B > Mo; in the fructification stage for *J. procumbens* and *J. tlaxcala*, they were Fe > Mn > Zn > B > Mo and Fe > Mn > B > Zn > Mo, respectively; in the harvesting stage for *J. procumbens*, it was Fe > Mn > B > Zn > Mo, and, for *J. tlaxcala*, Mn > Fe > B > Zn > Mo (Figure 7).

### 2.7. Total Dry Matter

The *J. procumbens* species compared to *J. tlaxcala* had a higher TDM for the vegetative, flowering, fructification, and harvesting stages by 35.5, 61.0, 97.4, and 162.7%, respectively. For EC, in flowering with 3 dS m^−1^, higher values were registered with respect to 1 dS m^−1^ by 65.2%; for the vegetative, fructification, and harvesting stages, there were no statistical differences, but the same trend was recorded: the higher EC, the higher TDM (Table 2).

### 2.8. Sps x EC Combination

The Sps x EC combination showed that the plant growth of both Sps with 3 dS m^−1^ EC favored the TDM in the vegetative stage (*J. procumbens* = 3.4 g plant^−1^ and *J. tlaxcala* = 2.7 g plant^−1^) (Appendix A), and in flowering, for *J. procumbens* in fructification and for *J. tlaxcala* in harvesting (Figure 8). For *J. tlaxcala* and *J. procumbens*, 2 dS m^−1^ EC favored them in the fructification and harvest stages.

### 2.9. Total Dry Matter Accumulation Order by Plant Structure

The distribution of dry matter in leaf (L), root (R), stem (S), and reproductive structures (RE) in *J. procumbens* was in the order of L ˃ R > S > RE in the vegetative and flowering stages, R > S > L > RE for fructification, and RE > R > S > L for harvesting. In *J. tlaxcala*, except for the accumulation of TDM in the vegetative stage which was equal to *J. procumbens* (L > R > S > RE), for the following three stages, a greater accumulation of TDM was registered in root and leaf (flowering = R > L > S > RE and fructification = R > L > RE > S) and root and reproductive structures (harvest = R > RE > L > S) (Figure 9).

## 3. Discussion

Variability in the macro- and micronutrients’ uptake was observed in both *Jaltomata* species. For macronutrients, *J. procumbens* absorbed a greater amount of N and P, while *J. tlaxcala* did so for Ca. N is involved in vegetative development. It is the most abundant macronutrient in the plant tissues, from 10 to 50 g kg^−1^ of total dry matter [18]. Its function is essential for the plant to complete its biological cycle [19]; it increases the photosynthetic production and duration in the leaf area, which are important characteristics because a larger leaf area is also a determinant of higher yields [20]; therefore, the greater uptake of this element is explained in the early stages of growth, where vegetative development is privileged, reducing uptake during fructification and harvesting to favor fruit development; the behavior of this is shown in both Sps in Figure 1. The N uptake rate is often positively correlated with K uptake, which is an element involved in cell expansion and stomatal opening, enzymatic activation, and osmotic adjustment, contributes up to 10% of the plant’s dry weight, and is the most abundant cation in plant cells [21]. This correlation is probably due to charge balance [19] or by the activation of the transporter involved in NO_3_^−^ and K^+^ assimilation, since recent works suggest that the NRT1.5 transporter, which directs the transport of NO_3_^−^ from the root to the stem, is also involved in the nitrate-dependent potassium translocation [22], and it is the main counterion for the translocation of NO_3_^−^ to the phloem [21].

Phosphorus had the same uptake dynamics in both Sps and constitutes between 0.5–5 g kg^−1^ of the dry weight of the plant, and the vacuole is the main storage site and plays an important role in cell division and carbohydrate metabolism [23], in the phosphorylation process of the NO_3_^−^ transporters, and in the low- and high-affinity transport systems NRT1 and NRT2 which regulate the absorption of NO_3_^−^ [22,24].

For Ca, the uptake rate and accumulation depend on the species [25]; however, the behavior observed in this work showed that both Sps had the same uptake dynamics (Figure 1). This element plays a structural and signaling role, and it can contribute to the osmotic balance; its concentration in plant tissue varies according to the crop conditions, organ and age of the plant, and the species [26]. This depends on the transpiration rate and water movement through the plant, and on the cationic exchange capacity of the cell wall, which influences the transport of Ca in the apoplast and its storage within the different tissues [25]; Ca is important for cell wall resistance and to prevent its enzymatic degradation, by contributing to the formation of bridges between pectins, resulting in Ca–pectate complexes, keeping fruit firmness and increasing their shelf life [27]. The relative distribution of total Ca is from 70 to 90% in leaves and from 10 to 30% in roots; its concentration in mature leaves can reach 10% of dry weight [26].

The average results of Mg in leaves were 3.5 and 3.6 g kg^−1^ for *J. tlaxcala* and *J. procumbens*, respectively, being in the upper limit of the optimal interval, 1.5–3.5 g kg^−1^, required in vegetative parts for plant growth [28], with the highest value in *J. tlaxcala*. Mg is important in the chlorophyll formation, it has a key role in the photosynthesis [29]; about 75% of Mg in the leaf is involved in protein synthesis and from 15 to 20% with chlorophyll pigments [28], because it is part of the complex of this molecule [30], whereby, it is preferably transported to this structure, and it also participates in the carbohydrate transport from the source organ to the demand organ [29]. This could respond to the greater absorption of Mg by *J. tlaxcala* at the fructification stage, as this element is required for the carbohydrate transport to the fruit, which represents the main organ of demand; for this Sps, a sucrose content in fruit of 9.4% is reported, while for *J. procumbens* it is 1.6% [10].

For micronutrients in both Sps, the B and Fe uptake presented a similar behavior, registering its lowest point in the flowering stage, and then its content increased. In both Sps, B registered its lowest point in flowering, which could indicate a greater distribution towards the flowers, where it plays a significant role in their development; subsequently, its concentration in the leaf increases, probably due to their greater requirement for the mobilization of sugars to the fruits. The B requirement in plants correlates with the pectin content [31], which could indicate a higher content of this polysaccharide in *J. tlaxcala*, by registering a higher concentration of B, than *J. procumbens*. The B plays an important role in the development of flowers, seeds, and fruit and participates in the transport of water, nutrients, and sugar towards growing regions; it is considered mobile through the phloem for many agricultural crops and transported as a complex with polyols; it is reported in plants from 10 to 200 mg kg^−1^ [32].

The Fe uptake depends on the species, genotype, environment, and age of the plant [33]; in plants, 140 mg kg^−1^ of dry weight is reported [34] and is stored in the vacuole where transporters such as VIT1 and IREG2 are in charge of importing the element into this organelle [33], where later it is removed to other organelles via the phloem, being the protein of the OPT3 oligopeptide transporter family, identified in Arabidopsis, the one involved in this process [35]. The Fe participates in the synthesis of DNA and chlorophyll and the reduction in N [36]; it is required for processes such as respiration and photosynthesis, by participating in the electron transport chain; it is a constituent of the heme portion of antioxidant enzymes such as catalases (CAT), non-specific peroxidases (POD), ascorbate peroxidases (APX), and as a cofactor of the Fe-superoxide dismutase [33].

Mn and Zn participate in the processes of photosynthesis and protein synthesis [37,38]. Mn also participates in respiration, enzyme activation, and synthesis of fatty acids, with a requirement in the plant of 20–40 mg kg^−1^ of dry weight [37] and is present in all the cellular compartments, being the vacuole which plays a role as temporary storage for its subsequent distribution to the other organelles, and its uptake and distribution is mediated by multiple transport proteins from diverse gene families [39]; on the other hand, Zn participates in other diverse physiological functions such as membrane structure and stress and diseases tolerance; its presence in plants is 30–100 mg kg^−1^ of dry matter [37].

For Mo, both Sps exceeded the requirement of this micronutrient (0.1–1.0 mg kg^−1^) with values for *J. procumbens* and *J. tlaxcala* of up to 4.5 and 3.65 mg kg^−1^, respectively. For some species with levels higher than 1.5 mg kg^−1^, anatomical alterations in the leaf, root, and stem are present; furthermore, it is reported that plants can tolerate high levels of up to 1000 mg kg^−1^ [34]. In this work, no negative effects on the plant development were observed; however, anatomical and physiological studies are required to determine the mechanisms of molybdenum sequestration and tolerance in both Sps at high levels of Mo uptake, as reported in *Brassica* species [40]. This element is important in the function of more than 50 different enzymes such as nitrogenase and nitrate reductase; the process of absorption and redistribution in the plant is still not entirely clear [41].

The effects of EC on the macro- and micronutrients’ uptake were observed. In elements such as Mg and Mn, the higher EC, the lower uptake of these nutrients was recorded, while the others, except for Ca in the last stage, showed an increase. This may respond to the fact that many plants have mechanisms to eliminate salts or to support their presence within their cells; so, high saline concentrations that could be harmful to some species are not harmful to others [42], which could be indicating a saline stress tolerance, involving an activation of gene modulations [43].

For SPAD readings, although the highest value was recorded at 2 dS m^−1^ EC, except for the vegetative stage, the highest SPAD readings were presented as EC increased, indicating a greater chlorophyll presence, because these readings are proportional to the concentration of this pigment in the leaf, where the available nutrients are important [7]. High chlorophyll concentrations have been found under conditions of high EC levels, reporting that the increase in chlorophyll under saline conditions improves stress tolerance [44].

For the dry matter accumulation in both Sps, the same dynamics were registered, with the greatest TDM increment for both Sps being from 45 dat, corresponding to the fructification stage (Figure 10), with a greater accumulation in *J. procumbens*, which is attributed to the characteristics of them, because *J. procumbens* presents a greater vegetative development with a higher plant height, thicker stem diameter, more ramifications, and more number of leaves [7]. In this work, *J. procumbens* exceeded *J. tlaxcala* in stem length (SL), root length (RL), stem diameter (SD), and leaf area (LA) in all the sampling stages (Appendix A), with values for the last sampling being the following: SL = 162.19 cm, RL = 53.73 cm, SD = 12.83 mm, and LA = 5980.20 cm^2^ plant^−1^, compared with *J. tlaxcala* which registered 64.36 cm for SL, 47.49 cm for RL, 7.91 mm for SD, and 2058.20 cm^2^ plant^−1^ for LA.

The importance of dry matter distribution is due to the yield of a crop, which is given by its ability to accumulate biomass in the organs destined for harvest [45].

In *J. procumbens*, the distribution of the TDM (vegetative and flowering = L ˃ R > S > RE; fructification = R > S > L > RE; and harvest = RE > R > S > L) would indicate that, in the early stages, this Sps focuses on the production of vegetative material, to generate the photosynthates that will be stored, and later translocated to the fruits, which represent the main sites of demand for the harvest period, competing among themselves and with the vegetative structures by available photoassimilates [45]. On the contrary, the distribution of TDM in *J. tlaxcala* (vegetative = L > R > S > RE; flowering = R > L > S > RE; fructification = R > L > RE > S; and harvest = R > RE > L > S) would indicate that this Sps spent less time accumulating dry matter in storage structures, allocating to the photoassimilates generated to supply the demand in the fruit development. This is due to the nature of both genotypes because *J. procumbens* presents a continuous and superior generation of ramifications, inflorescences, and fruits than *J. tlaxcala*; therefore, it requires a greater accumulation of reserves to supply photoassimilates to the sites with the greatest demand in the fructification and harvesting stages, which also explains the smaller size and weight of the fruit reported for *J. procumbens* [7], since the greater number of fruits per plant, the smaller the dry matter partition to each of these [46].

As mentioned previously, these species are found in traditional food production systems, with particular environmental and soil conditions depending on the site where they are established. Therefore, it is important to use this information to carry out fertilization to first approximate the different soil conditions where these species grow naturally and what strategies can be applied to maintain their populations.

## 4. Materials and Methods

The research was carried out in a greenhouse at the Colegio de Postgraduados, Campus Montecillo, Texcoco, State of Mexico, from August to November 2020. Water and nutrients were supplied through drip irrigation without nutrient-solution reuse.

### 4.1. Genetic Material

The species *Jaltomata procumbens* of an erect growth habit and blue dark fruit and *Jaltomata tlaxcala* of a decumbent growth with green fruit in the mature stage, from Tlaxcala, Mexico (19° 15′ N and 97° 53′ W, at 2500 m of altitude), were evaluated; the region has a predominant temperate subhumid climate C(w1) and C(w2) and an average annual temperature between 12 °C and 18 °C. Seeds of the fruits obtained from a previous work in 2019 were used. The seeds were extracted and rinsed off with tap water, they were dried under shadow at room temperature (22 °C), and they were stored in paper bags of 125 g capacity and maintained in refrigerated conditions at 4 °C.

### 4.2. Experiment Management

The seeds were placed in petri dishes and were soaked with distilled water for four days and maintained in a germination chamber (model: ATTGPT-B; series: 143958201) with diurnal (12 h at 30 °C ± 1) and nocturnal (12 h at 20 °C ± 1) conditions. When necessary, distilled water was used to keep the seeds under soaking conditions.

After soaking time, the seeds were extracted and placed in a sieve (mesh number 22) and rinsed with distilled water for 10 s. The seeds were distributed on moistened filter paper in petri dishes and kept in the germination chamber under the aforementioned conditions. Distilled water was applied to keep the paper moistened.

The germinated seeds, with cotyledonous leaves, were put in plastic cups of 24 mL capacity with peat moss as a substrate and irrigated with distilled water. They were kept for two days in the germination chamber; after that, these were placed in a greenhouse with 40% shade cloth and irrigated with tap water (pH 7.7 and 0.5 dS m^−1^ EC). Then, the plants were transplanted in 265 mL Styrofoam cups with red porous gravel locally known as tezontle as a substrate (particle diameter of 1 to 3 mm) and irrigated with a Steiner nutrient solution with 1 dS m^−1^ EC until they reached a high of 15 cm. Forty-two days after sowing, the plants were transplanted to 40 × 40 black polyethylene bags (13 L) with red tezontle as a substrate (diameter of ≤12 mm, 0.78 g cm^−3^ density, and 21.9% humidity retention). The main stem and branches were moored during the plant growing. In *J. procumbens*, lateral and basal pruning were performed, and in *J. Tlaxcala*, two basal shoots were kept, and the lateral shoots were eliminated. Pests and pathogens, such as beet armyworm (*Spodoptera exigua*) and powdery mildew (*Oidium* sp.), were presented. Beet armyworms were eliminated manually, whereas for powdery mildew, sodium bicarbonate (20 g L^−1^) was applied.

Four destructive samples were taken at 17, 31, 45 and 57 dat: vegetative, flowering, fructification, and harvesting stages, respectively. The sampled plants were extracted from the substrate and the root was washed with tap water, and at the laboratory the root, leaves, stem, and reproductive structures were separated. To obtain the dry weights, each sample was placed in paper bags and dried in a stove for 72 h (Riossa Stove, model HCF-125D, series 120209).

### 4.3. Experimental Design

A factorial design was established with the next factors and levels: two species (*J. procumbens* and *J. tlaxcala*), three concentrations (EC of nutrient solution) of Steiner nutrient solution [47], with macro- and micronutrients supplied proportionally to each EC level: 1, 2, and 3 dS m^−1^, dosed from the transplant; the irrigation volume applied daily was from 0.23 to 1 L per plant, depending on the growth stage. The pH was between a 5.5 to 6.5 interval. The pH and EC were monitored using a portable equipment (Conductronic PC18). The treatments were distributed in a completely randomized design with three replications. The experimental unit was one plant per pot, with 72 experimental units in total.

### 4.4. Variables Evaluated

#### 4.4.1. Macro- and Micronutrients Analysis in Leaves

The dried leaf samples were milled and passed through a mesh number 40. Then 0.25 g was weighed, and a humid digestion was carried out using a mix of H_2_SO_4_ and HClO_4_ (2:1, *v*/*v*), with 1 mL of H_2_O_2_ at 30%. After that, each sample was diluted to 25 mL with deionized water and then filtered. Except for N, all the nutrients were determined with a coupled plasma atomic emission spectrometry equipment (ICP-AES 725-ES, Agilent, Santa Clara, CA, USA). The N were determined with the Semimicro-Kjeldahl method [48]. The results are reported in g kg^−1^ for macronutrients and in mg kg^−1^ for micronutrients.

#### 4.4.2. Plant Characteristics

Before each sampling, the SPAD-502 readings were used to measure the intensity of the green color of the leaves. An average value of four recently expanded opposite leaves was obtained.

Stem height (SH) and root length (RL) were registered with a tape measure in cm, the stem diameter (SD) with a vernier caliper in mm, and the leaf area (LA) with a leaf area integrator in cm^2^ (LI-COR, Inc. Lincoln, NE, USA, LI-3100 Area meter). The total dry matter (TDM) was obtained with the sum of the dry weights of each organ and was measured with an analytical balance in g (Adventurer Pro AV213C Model).

#### 4.4.3. Statistical Analysis

A variance analysis was performed based on a completely random factorial model 2 × 3 and a Tukey’s separation means test (*p* ≤ 0.05). The SAS program (2008) 9.2 version statistical program was used [49].

## 5. Conclusions

A variability was registered between the species and phenological stages, in the macro- and micronutrients concentration, and in the total dry matter accumulation.

The *Jaltomata procumbens* absorbed a higher amount of N, P, and Mo, while *Jaltomata tlaxcala* absorbed more Ca, B, Fe, and Zn.

The plant growth of both species with high electrical conductivity favored the total dry matter accumulation.

Due to this variability, specific fertilization programs are required for each species. The optimal nutrient absorption and biomass production are obtained with the nutrient solutions with 2 and 3 dS m^−1^ EC, for *Jaltomata procumbens* and *Jaltomata tlaxcala*, respectively, in a greenhouse and using hydroponics.

It is necessary to use these data to carry out fertilization to first approximate the field conditions due to the different soil characteristics where these species grow.

## Figures and Tables

**Figure 1 plants-14-01124-f001:**
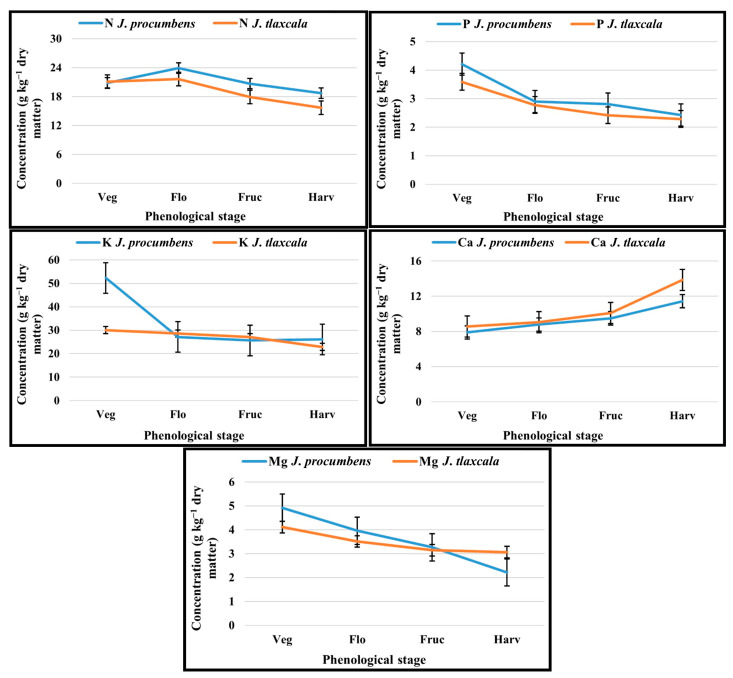
Uptake of N, P, K, Ca, and Mg in *Jaltomata procumbens* and *Jaltomata tlaxcala*. Colegio de Postgraduados, Campus Montecillo, Texcoco, State of Mexico. September 2020–February 2021. The samplings were carried out at 17, 31, 45, and 57 days after transplant (dat), corresponding to the stages: Veg, Vegetative; Flo, Flowering; Fruc, Fructification; and Harv, Harvesting. DMSH: Stage: vegetative N (3.22 g kg^−1^), P (0.31 g kg^−1^), K (5.76 g kg^−1^), Ca (0.68 g kg^−1^), Mg (0.49 g kg^−1^); flowering N (1.99 g kg^−1^), P (0.19 g kg^−1^), K (2.78 g kg^−1^), Ca (0.81 g kg^−1^), Mg (0.40 g kg^−1^); fructification N (1.51 g kg^−1^), P (0.24 g kg^−1^), K (1.14 g kg^−1^), Ca (0.67 g kg^−1^), Mg (0.26 g kg^−1^); harvesting N (1.47 g kg^−1^), P (0.42 g kg^−1^), K (3.01 g kg^−1^), Ca (1.38 g kg^−1^), Mg (0.43 g kg^−1^) (Tukey, *p* ≤ 0.05).

**Figure 2 plants-14-01124-f002:**
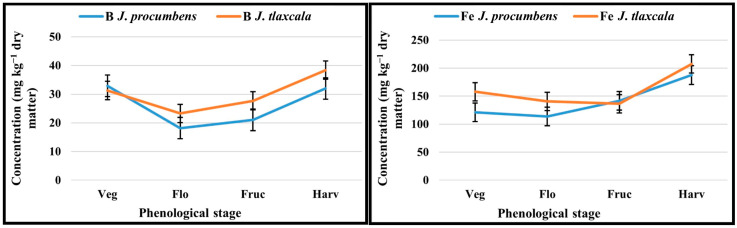
Uptake of B, Fe, Mn, Mo, and Zn in *Jaltomata procumbens* and *Jaltomata tlaxcala*. Colegio de Postgraduados, Campus Montecillo, Texcoco, State of Mexico. September 2020–February 2021. The samplings were carried out at 17, 31, 45, and 57 dat, corresponding to the stages: Veg, Vegetative; Flo, Flowering; Fruc, Fructification; and Harv, Harvesting. DMSH: Stage: vegetative B (13.34 mg kg^−1^), Fe (22.14 mg kg^−1^), Mn (9.76 mg kg^−1^), Mo (1.60 mg kg^−1^), Zn (7.00 mg kg^−1^); flowering B (2.19 mg kg^−1^), Fe (33.11 mg kg^−1^), Mn (10.47 mg kg^−1^), Mo (1.91 mg kg^−1^), Zn (4.22 mg kg^−1^); fructification B (3.53 mg kg^−1^), Fe (25.58 mg kg^−1^), Mn (14.43 mg kg^−1^), Mo (1.92 mg kg^−1^), Zn (3.24 mg kg^−1^); harvest B (6.08 mg kg^−1^), Fe (59.27 mg kg^−1^), Mn (36.20 mg kg^−1^), Mo (2.39 mg kg^−1^), Zn (4.10 mg kg^−1^) (Tukey, *p* ≤ 0.05).

**Figure 3 plants-14-01124-f003:**
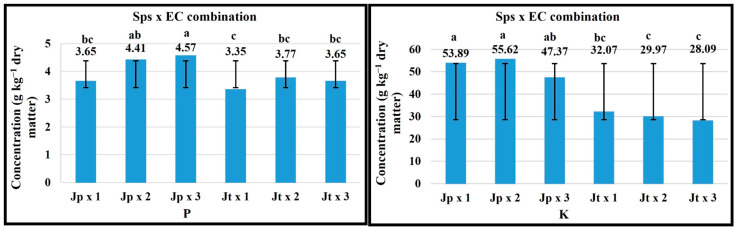
Effect of the Sps x EC combination on the concentration of K, P, Ca, Mg, and Fe in vegetative stage of *Jaltomata procumbens* and *Jaltomata tlaxcala*. Abbreviations: Jp × 1, *J. procumbens* x 1 dS m^−1^; Jp × 2, *J. procumbens* × 2 dS m^−1^; Jp × 3, *J. procumbens* × 3 dS m^−1^; Jt × 1, *J. tlaxcala* × 1 dS m^−1^; Jt × 2, *J. tlaxcala* × 2 dS m^−1^; Jt × 3, *J. tlaxcala* × 3 dS m^−1^. Colegio de Postgraduados, Campus Montecillo, Texcoco, State of Mexico. September 2020–February 2021. Data taken between 17 and 129 dat. Different letters among bars indicate statistical difference (Tukey, *p* ≤ 0.05).

**Figure 4 plants-14-01124-f004:**
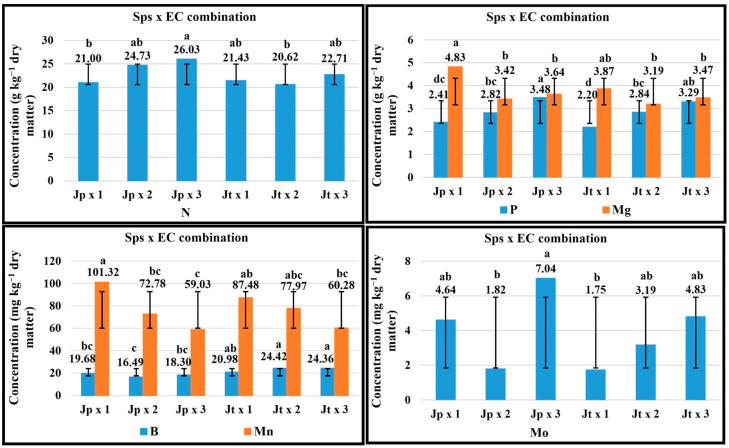
Effect of the Sps x EC combination on the concentration of N, P, Mg, Bo, Mn, and Mo in flowering stage of *Jaltomata procumbens* and *Jaltomata tlaxcala*. Abbreviations: Jp × 1, *J. procumbens* x 1 dS m^−1^; Jp × 2, *J. procumbens* x 2 dS m^−1^; Jp × 3, *J. procumbens* × 3 dS m^−1^; Jt × 1, *J. tlaxcala* × 1 dS m^−1^; Jt × 2, *J. tlaxcala* × 2 dS m^−1^; Jt × 3, *J. tlaxcala* × 3 dS m^−1^. Colegio de Postgraduados, Campus Montecillo, Texcoco, State of Mexico. September 2020–February 2021. Data taken between 17 and 129 dat. Different letters among bars indicate statistical difference (Tukey, *p* ≤ 0.05).

**Figure 5 plants-14-01124-f005:**
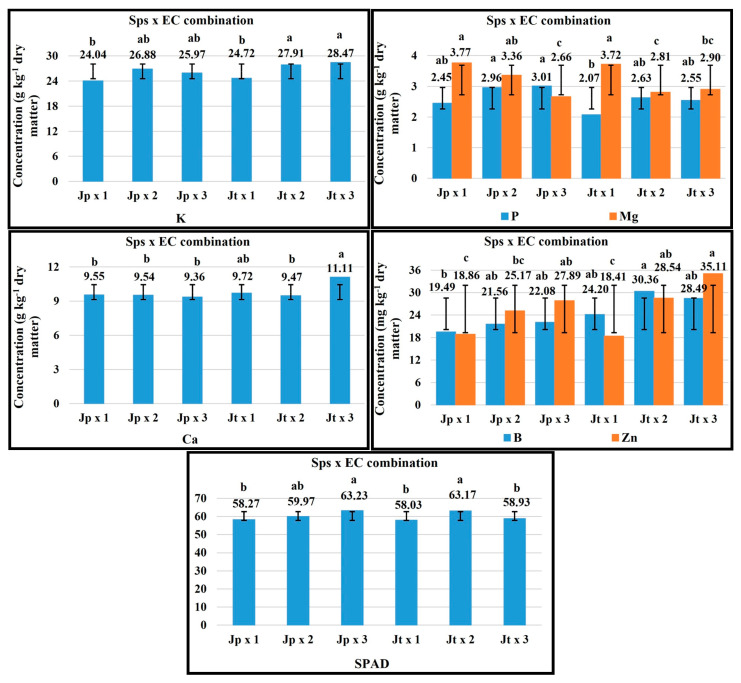
Effect of the Sps x EC combination on the concentration of K, P, Ca, Mg, B, and Zn and SPAD readings, in fructification stage of *Jaltomata procumbens* and *Jaltomata tlaxcala*. Abbreviations: Jp × 1, *J. procumbens* x 1 dS m^−1^; Jp × 2, *J. procumbens* × 2 dS m^−1^; Jp × 3, *J. procumbens* × 3 dS m^−1^; Jt × 1, *J. tlaxcala* × 1 dS m^−1^; Jt × 2, *J. tlaxcala* × 2 dS m^−1^; Jt × 3, *J. tlaxcala* × 3 dS m^−1^. Colegio de Postgraduados, Campus Montecillo, Texcoco, State of Mexico. September 2020– February 2021. Data taken between 17 and 129 dat. Different letters among bars indicate statistical difference (Tukey, *p* ≤ 0.05).

**Figure 6 plants-14-01124-f006:**
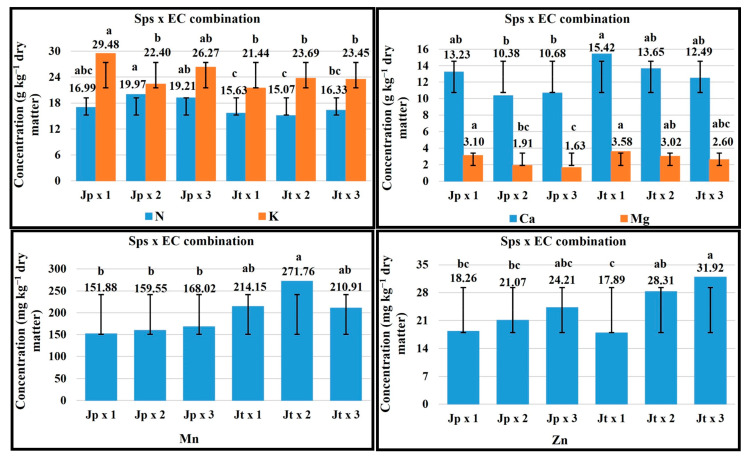
Effect of the Sps x EC combination on the concentration of the N, K, Ca, Mg, Mn, and Zn, in harvest stage of *Jaltomata procumbens* and *Jaltomata tlaxcala*. Abbreviations: Jp × 1, *J. procumbens* × 1 dS m^−1^; Jp × 2, *J. procumbens* × 2 dS m^−1^; Jp × 3, *J. procumbens* × 3 dS m^−1^; Jt × 1, *J. tlaxcala* × 1 dS m^−1^; Jt × 2, *J. tlaxcala* × 2 dS m^−1^; Jt × 3, *J. tlaxcala* × 3 dS m^−1^. Colegio de Postgraduados, Campus Montecillo, Texcoco, State of Mexico. September 2020–February 2021. Data taken between 17 and 129 dat. Different letters among bars indicate statistical difference (Tukey, *p* ≤ 0.05).

**Figure 7 plants-14-01124-f007:**
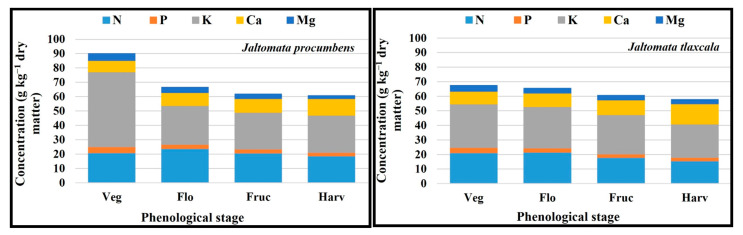
Order of the uptake level on the macronutrients and micronutrients of *Jaltomata procumbens* and *Jaltomata tlaxcala*. Colegio de Postgraduados, Campus Montecillo, Texcoco, State of Mexico. September 2020–February 2021. Sampling carried out at 17, 31, 45, and 57 dat, corresponding to the stages: Veg, Vegetative; Flo, Flowering; Fruc, Fructification; Harv, Harvesting.

**Figure 8 plants-14-01124-f008:**
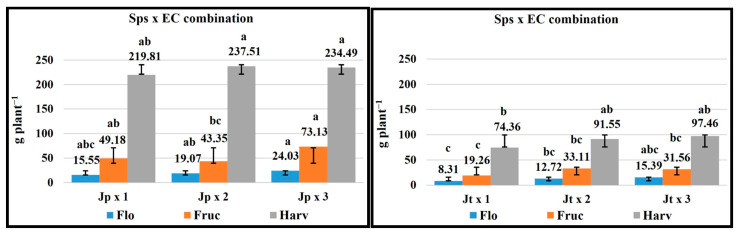
Effect of the Sps × EC combination on the total dry matter of *Jaltomata procumbens* and *Jaltomata tlaxcala*. Abbreviations: Jp × 1, *J. procumbens* × 1 dS m^−1^; Jp × 2, *J. procumbens* × 2 dS m^−1^; Jp × 3, *J. procumbens* × 3 dS m^−1^; Jt × 1, *J. tlaxcala* × 1 dS m^−1^; Jt × 2, *J. tlaxcala* × 2 dS m^−1^; Jt × 3, *J. tlaxcala* × 3 dS m^−1^. Colegio de Postgraduados, Campus Montecillo, Texcoco, State of Mexico. September 2020–February 2021. Data taken between 17 and 129 dat. Different letters among bars indicate statistical difference (Tukey, *p* ≤ 0.05).

**Figure 9 plants-14-01124-f009:**
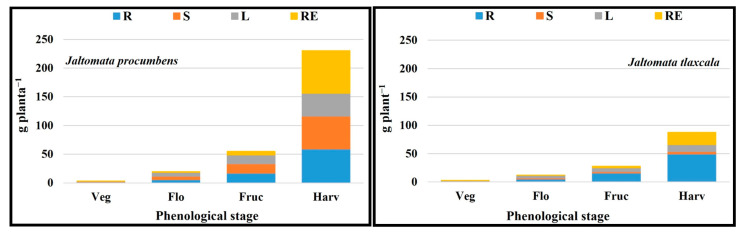
Dry matter accumulation by leaf (L), stem (S), root (R), and reproductive structures (RE) in the vegetative (Veg), flowering (Flo), fructification (Fruc), and harvesting (Harv) stages of *Jaltomata procumbens* and *Jaltomata tlaxcala*. Colegio de Postgraduados, Campus Montecillo, Texcoco, State of Mexico. September 2020–February 2021. Data taken between 17 and 129 dat.

**Figure 10 plants-14-01124-f010:**
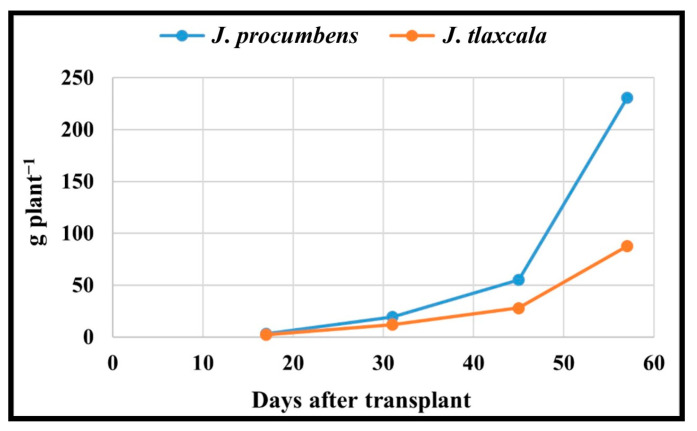
Dynamics of dry matter accumulation of *Jaltomata procumbens* and *Jaltomata tlaxcala*. Colegio de Postgraduados, Campus Montecillo, Texcoco, State of Mexico. September 2020–February 2021. Data taken between 17 and 129 dat.

**Table 1 plants-14-01124-t001:** Comparison of means on the concentration of N, P, K, Ca, Mg, Fe, Mn, Mo, and Zn of *Jaltomata procumbens* and *Jaltomata tlaxcala* cultivated at three electrical conductivity levels of nutrient solution. Colegio de Postgraduados, Campus Montecillo, Texcoco, State of Mexico. September 2020–February 2021. Data taken between 17 and 129 days after transplant (dat).

FV	Macronutrients (g kg^−1^ Dry Matter)					Micronutrients (mg kg^−1^ Dry Matter)		
EC (dS m^−1^)	Vegetative Stage (17 dat)					Vegetative Stage (17 dat)		
	N	P		Ca	Mg	Fe		Mo
1	17.8 b	3.5 b		7.4 b	4.8 a	118.3 b		1.9 b
2	21.6 ab	4.1 a		8.7 a	4.7 a	146.6 ab		4.1 a
3	23.2 a	4.1 a		8.6 a	4.0 b	153.4 a		5.2 a
	Flowering stage (31 dat)					Flowering stage (31 dat)		
	N		P		Mg	Mn	Mo	Zn
1	21.2 b		2.3 c		4.3 a	94.4 a	3.2 b	24.2 b
2	22.6 ab		2.8 b		3.3 b	75.4 b	2.5 b	25.8 ab
3	24.3 a		3.4 a		3.6 b	59.7 c	5.9 a	30.0 a
	Fructification stage (45 dat)					Fructification stage (45 dat)		
	P	K			Mg	Mn	Mo	Zn
1	2.3 b	24.4 b			3.7 a	118.3 a	1.6 b	18.6 c
2	2.8 a	27.4 a			3.1 b	112.0 ab	4.8 a	26.9 b
3	2.8 a	27.2 a			2.8 b	94.7 b	4.3 a	31.5 a
	Harvesting stage (57 dat)					Harvesting stage (57 dat)		
	Ca			Mg		Zn		
1	14.3 a			3.3 a		18.1 b		
2	12.0 b			2.5 b		24.7 a		
3	11.6 b			2.1 b		28.1 a		

Abbreviations: dat, days after transplant; EC, electrical conductivity. Different letters indicate statistical difference (Tukey, *p* ≤ 0.05).

**Table 2 plants-14-01124-t002:** Comparison of total dry matter means of *Jaltomata procumbens* and *Jaltomata tlaxcala*, cultivated at three electrical conductivity levels of nutritive solution. Colegio de Postgraduados, Campus Montecillo, Texcoco, State of Mexico. September 2020–February 2021. Data taken between 17 and 129 dat.

FV	Total Dry Matter (g plant^−1^)			
Species	Vegetative	Flowering	Fructification	Harvesting
*Jaltomata procumbens*	3.1 a	19.6 a	55.2 a	230.6 a
*Jaltomata tlaxcala*	2.3 b	12.1 b	28.0 b	87.8 b
EC (dS m^−1^)				
1	2.4 a	11.9 b	34.2 a	147.1 a
2	2.7 a	15.9 ab	38.2 a	164.5 a
3	3.0 a	19.7 a	52.4 a	165.9 a

Abbreviation: EC, electrical conductivity. Different letters indicate statistical differences (Tukey, *p* ≤ 0.05).

## Data Availability

The original contributions presented in this study are included in the article/Appendix A. Further inquiries can be directed to the corresponding author(s).

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
