# Peer review of "Nutrient Uptake of Two Semidomesticated *Jaltomata* Schltdl. Species for Their Cultivation"

_plants, 2025, doi:10.3390/plants14071124_

Round 1
Reviewer 1 Report
Comments and Suggestions for Authors
I want to congratulate you for this interesting article.

Author Response
|
1. Summary |
|
|
|
Thank you very much for taking the time to review this manuscript. Please find the detailed responses below and the corresponding revisions/corrections highlighted/in track changes in the re-submitted files.
|
||
|
2. Questions for General Evaluation |
Reviewer’s Evaluation |
Response and Revisions |
|
Does the introduction provide sufficient background and include all relevant references? |
Yes |
[Please give your response if necessary. Or you can also give your corresponding response in the point-by-point response letter. The same as below] |
|
|
|
|
|
Is the research design appropriate? |
Yes |
|
|
Are the methods adequately described? |
Yes |
|
|
Are the results clearly presented? |
Yes |
|
|
Are the conclusions supported by the results?
|
Yes |
|
|
3. Point-by-point response to Comments and Suggestions for Authors |
||
|
Comments 1: Abstract. Write where the study was carried out and the period of experimentation. Some recorded macro and micronutrient concentration values may also be included.
|
||
|
Response 1: Thank you for pointing this out. I agree with this comment. The recommendations were addressed, the place of the study was carried out and the period of experimentation, and macro and micronutrient concentration values were added. Page number: 1; paragraph: 1; line: 14-23
|
||
|
Comments 2: Materials and Methods. I recommend entering these part immediately after Line 65 (2. Materials and Methods). Write some distinct characteristics that differentiate the two species studied. |
||
|
Response 2: Thank you for pointing this out about entering this part immediately after Line 65, but the document is written taking into account the Plants journal template. We are not sure if it could be changed
As for the distinct characteristics that differentiate the two species studied, they were added. Page number: 14; paragraph: 6; line: 432
Comments 3: The numbering will change throughout the manuscript (2. Results will be written 3. Results; 3. Discussions will be written 4. Discussions etc... ).
Response 3: Thank you for pointing this out about changing the manuscript numbering, but the document is written taking into account the Plants journal template. We are not sure if it could be changed.
|
||
|
|
||

Reviewer 2 Report
Comments and Suggestions for Authors
The purpose of this study was to determine nutrient uptake curves for J. procumbens and J. tlaxcala in greenhouse and hydroponics at different phenological stages (vegetative, flowering, fruiting and harvesting).
-The results can be used to develop optimal fertilization strategies to increase the growth and yield of these plants, especially in the cultivation of Jaltomata procumbens and Jaltomata tlaxcala under controlled conditions such as greenhouses and hydroponics
- However, the authors did not address the impact of environmental stress, genotype-environment interactions, and soil and climate specifics, resulting in a major news deficiency.
- The authors should make additions to the description of the work and expand on the anatomical studies that should be conducted to determine the behavior of both Jaltomata species at high levels of molybdenum uptake (Mo.).
- In addition, the authors could expand the discussion on the possibility of relating the obtained results to field conditions, taking into account the peculiarities of these conditions and the need to adapt fertilization programs.
Comments on the Quality of English LanguageThe language is of good quality.
Author Response
|
1. Summary |
|
|
|
Thank you very much for taking the time to review this manuscript. Please find the detailed responses below and the corresponding revisions/corrections highlighted/in track changes in the re-submitted files.
|
||
|
2. Questions for General Evaluation |
Reviewer’s Evaluation |
Response and Revisions |
|
Does the introduction provide sufficient background and include all relevant references? |
Yes |
[Please give your response if necessary. Or you can also give your corresponding response in the point-by-point response letter. The same as below] |
|
|
|
|
|
Is the research design appropriate? |
Can be improved |
|
|
Are the methods adequately described? |
Can be improved |
|
|
Are the results clearly presented? |
Yes |
|
|
Are the conclusions supported by the results?
|
Can be improved |
|
|
3. Point-by-point response to Comments and Suggestions for Authors |
||
|
Comments 1: The authors did not address the impact of environmental stress, genotype-environment interactions, and soil and climate specifics, resulting in a major news deficiency.
|
||
|
Response 1: Thank you for pointing this out. These characteristics are important, however they were not considered because it was a research under controlled conditions.
|
||
|
Comments 2: The authors should make additions to the description of the work and expand on the anatomical studies that should be conducted to determine the behavior of both Jaltomata species at high levels of Molybdenum uptake (Mo). |
||
|
Response 2: Thank you for pointing this out. Recommendations were taken into account. Page number: 13; paragraph: 2; line: 370-372
Comments 3: The authors could expand the discussion on the possibility of relating the obtained results to field conditions, taking into account the peculiarities of these conditions and the need to adapt fertilization programs.
Response 3: Thank you for pointing this out. It was a first approximation to determine the nutrient requirements of both species, because these species are semi-domesticated and are more efficient in the use of nutrients, but we added information about the characteristics of the traditional production systems in Mexico, and the importance of using the information to carry out fertilizations for field conditions.
Page numbers: 1, 14 and 16; paragraphs: 2, 4 and 8; lines: 40-44; 421-425 and 518-519. |
||
